# Mixed-methods evaluation and behavior change interventions to improve hand hygiene resources and practices among healthcare workers in polyclinics and health centers in Belize, 2023

Anh N. Ly[1], Kelsey McDavid[2], Christina Craig[2], Rosalva Blanco[1], Vickie Romero[1], Melissa Diaz-Musa[3], Francis Morey[3], Russell Manzanero[3], Gerhaldine Morazan[1], Makenzie Towery[2], Anna Impellitteri[2], Matthew Lozier[2,4], Kristy O. Murray[1,5]*

1 Department of Pediatrics, National School of Tropical Medicine, Baylor College of Medicine and Texas Children's Hospital, Houston, Texas, United States of America, 2 Division of Foodborne, Waterborne, and Environmental Diseases, Centers for Disease Control and Prevention, Atlanta, Georgia, United States of America, 3 Belize Ministry of Health and Wellness, Belmopan, Belize, 4 United States Public Health Service, Silver Springs, Maryland, United States of America, 5 Emory University and Children's Healthcare of Atlanta, Atlanta, Georgia United States of America

☯ These authors contributed equally to this work.
* kristy.murray@emory.edu

**Data availability statement:** All relevant data are within the manuscript.

## Abstract

### Background

Hand hygiene (HH) is an effective public health measure to prevent the spread of infections in healthcare settings. A previous study in Belize showed gaps in HH practices in hospitals and large polyclinics; however, there are limited national data assessing access to and use of HH resources in smaller outpatient primary care facilities, especially in rural areas.

### Methods

In February 2023, facility assessments were conducted at 26 health centers and polyclinics in Belize to assess the availability of HH resources. Of these, 12 pilot healthcare facilities (HCF) were selected for additional evaluation, which included observation of HH practices, hand dirtiness assessments, and in-depth interviews. Following the evaluation, a six-week (August – September 2023) hand hygiene champion intervention was implemented at the 12 pilot HCFs to promote HH practices. Follow-up assessments were conducted during September – November 2023 to evaluate the impact of the intervention. Descriptive statistics and adjusted odds ratios were calculated to assess HH resources, adherence, and dirtiness. Thematic analysis was conducted for the in-depth interviews.

**Funding:** This project is funded through a cooperative agreement with the US Centers for Disease Control and Prevention (U01 GH002235, PI: Kristy O. Murray). Additional funding support was provided by USAID.

**Competing interests:** The authors have declared that no competing interests exist.

## Results

Most (87%) patient care rooms at the HCFs had either a handwashing station with soap and water or a functional alcohol-based hand rub dispenser. Following the intervention, there was a significant increase in hand hygiene adherence (HHA) among healthcare workers (aOR = 4.21; 95% CI = 2.70, 6.56). Overall, HHA was more common during invasive procedures (aOR = 1.82; 95% CI = 1.07, 3.09) and after patient contact (aOR = 1.68; 95% CI = 1.12, 2.52). The median hand dirtiness score increased from 8 to 9, indicating less visible debris. In-depth interviews found that healthcare workers viewed the intervention as a helpful reminder but encountered challenges such as having few staff, lack of time, and lack of resources during program implementation.

## Conclusion

The observed increase in HHA and positive feedback from healthcare workers suggests that a peer-led program may be an effective strategy to improve HHA in HCFs. Future programs may consider tailoring the intervention to the resource and adherence gaps observed at each facility to increase impact.

## Introduction

Hand hygiene (HH) is an important infection prevention and control (IPC) measure to reduce nosocomial infections in healthcare settings, especially in low- and middle-income countries where higher rates of nosocomial infections occur [1]. However, there are limited data on hand hygiene adherence (HHA) in low- and middle-income countries. Furthermore, despite the importance of HH in healthcare facilities (HCFs), adherence among healthcare workers (HCWs) varies. A multi-country study found that overall HCW pre-intervention HHA ranged from 22.4% in low-and-middle-income countries to 54.3% in high-income countries [2]. A meta-analysis showed that HHA was 74% during the COVID-19 pandemic, which was higher than previously observed [3]. Globally, studies and IPC programs have focused more on larger HCFs with inpatient care services. However, IPC measures are critical in outpatient settings where patients most often present during the early stage of clinical illness. A study conducted in outpatient wound care clinics in the United States found that HCWs acquired at least one healthcare-associated pathogen on their hands during 28.3% of patient care encounters [4].

Our study took place in Belize, an upper-middle-income country in Central America, bordering Mexico, Guatemala, and the Caribbean Sea. In Belize, public polyclinics and health centers provide outpatient services and serve as the first level of care for the local communities, many of which are in rural areas of the country. To our knowledge, there is no prior published assessment of HH resources and practices in these HCFs. Our team conducted a similar study at the 11 largest public HCFs (including hospitals) in Belize to assess HH resources and practices in 2021–2022 [5]. This study expanded the evaluation to 26 smaller health centers and outpatient

polyclinics that were not previously included. Since these health centers and polyclinics are limited in size, clinical capacity, and resources, HH practices and needs may differ from those observed at larger HCFs.

This study aims to assess HH resources and evaluate the impact of a behavior change intervention on HHA among HCWs in health centers and polyclinics in Belize. We assessed hand hygiene resources at 26 HCFs and conducted a pre/post-intervention evaluation of HHA in 12 pilot HCFs.

## Methods

### Study design

A facility assessment was conducted at 26 public polyclinics and health centers in Belize to understand access to hand hygiene services at the national level. These 26 sites were selected from 53 public polyclinics and health centers in Belize since they are a) open for service at the start of the study and b) not part of our previous hand hygiene study. Of these 26 facilities, 12 were selected for an additional in-depth evaluation and pilot intervention (these 12 will be referred to as pilot HCFs). These 12 pilot HCFs were selected prior to the facility assessment based on staff size as a proxy for the population utilizing the services. In each of the six districts in the country, the two facilities with the highest number of staff were selected to maximize the impact of the evaluation and intervention. In three districts, multiple facilities had the same number of staff; therefore, the pilot facilities for the district were randomly selected.

In addition to the facility assessment, observations of HH practices among HCWs during patient contact, hand dirtiness assessments among HCWs who have patient contact during the workday, and an in-depth interview with staff were conducted in the 12 pilot HCFs at baseline and follow-up (Fig 1). To reduce bias in the data, the assessments were conducted in the following order at each facility: HH observations, hand dirtiness assessment, facility assessment, and in-depth interviews. The baseline assessments were conducted in-person in from February 1, 2023 to February 16, 2023. The intervention was implemented during August – September 2023, and follow-up assessments were conducted from September 27, 2023 to November 2, 2023. This study was approved by the Ethics Committee at the Belize Ministry of Health and Wellness and the institutional review board at Baylor College of Medicine (BCM) (protocol H-49250). The study was also reviewed by the U.S. Centers for Disease Control and Prevention (CDC), deemed not research, and was conducted consistent with applicable federal law and CDC policy (45 CFR 46.102(l)).

### Facility assessment

The facility assessments were conducted by trained BCM enumerators. In each patient care area, the enumerators documented the presence and functionality of HH resources such as handwashing stations, soap, and alcohol-based hand rub (ABHR) (excluding personal or surplus ABHR supplies). A functional ABHR dispenser was defined as a dispenser with ABHR inside and mechanically operating. Facility assessments were documented on REDCap using tablets [6,7].

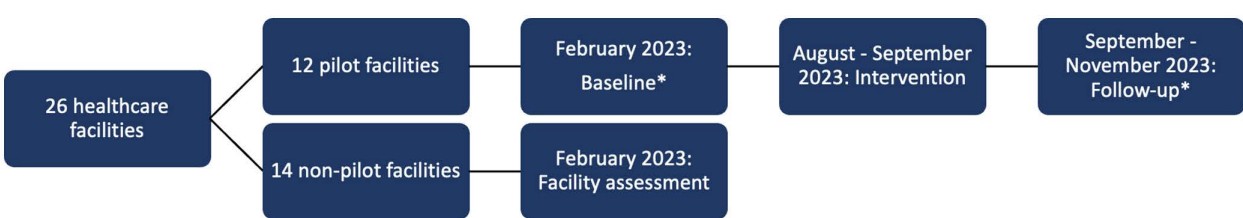

*Data collection included facility assessment, hand hygiene observations, hand dirtiness assessment, and in-depth interviews

**Fig 1. Study timeline and components.**

Although the intervention in the 12 pilot HCFs did not include the provision of HH supplies, a pre-post evaluation was conducted at these HCFs to capture any changes in the availability of HH resources between the baseline and follow-up assessments since access to resources may influence practices. This comparison was necessary because procurement of supplies at these facilities may have changed based on the prioritization of HH supplies at the national and local levels as the COVID-19 pandemic progressed. The pre-post assessment of HH resources also allows us to evaluate whether the behavior change intervention facilitated an increase in the availability of HH resources.

## Hand hygiene observations

Observations were conducted during provider-patient interactions to document HCWs' HH practices before and after patient contact at the 12 pilot HCFs. The observation tool was adapted from the World Health Organization's (WHO) 5 Moments for Hand Hygiene [8]. All observations were conducted by trained Belizean BCM enumerators. At each HCF, a total of three HCWs who had physical contact with patients were randomly selected for observation. For each HCW, the enumerators were to observe up to five patient contacts or up to one hour, whichever occurred first. For each patient contact, the following information was recorded: HCW occupation, contact with new patient or repeated contact, HH materials present, type of patient contact (invasive vs. non-invasive), and HH method used before and after patient contact. Procedures that involved contact with broken skin or mucous membranes were classified as invasive. A patient contact started when the HCW had any physical contact with the patient and ended when the patient or the HCW left the room or when the HCW touched any other item that is not a medical tool (e.g., cell phone, desk drawer, door handle, etc.). For each patient consultation, multiple patient contacts can occur. HH observations were only conducted in rooms where there was water with/without soap or ABHR. HHA was defined as washing hands with soap and water or using ABHR. Note that since HHA was recorded before and after each patient contact, certain instances of the WHO's 5 Moments for Hand Hygiene may have been missed (e.g., moving from body fluid exposure to aseptic procedures on the same patient). To minimize bias, the enumerators informed the HCWs that they were there to observe the different types of procedures performed or the quality of care to prevent influence on HH behaviors. No personal or health information was collected from the HCWs or their patients. Verbal consent was obtained from all HCWs before the observations. Observations were documented on paper and then transferred to REDCap [6,7]. Some HCWs may have participated at both baseline and follow-up, but the observations were not paired.

## Hand dirtiness assessment

A hand dirtiness assessment was conducted using the Quantitative Personal Hygiene Assessment Tool (qPHAT) [9]. All HCWs present on the day of the assessment who normally have physical contact with patients were invited to participate. Verbal consent was obtained from each participant. A sterile saline gauze (Hygea brand) was used to trace along the peripheral and diagonal section of the right palm and each finger pad of the right hand of each participant. The gauze was scored at the HCFs by a trained enumerator. The densest half-inch square of the gauze was compared to the color scale below (Fig 2) to determine the hand dirtiness score. If the half-inch square was between two values, the lower score was reported. A higher score corresponds to less visible debris. To assess HH knowledge, each participant was asked the following question: "When your hands are visibly dirty, should you clean them with soap and water, alcohol-based hand sanitizer, or are both equally effective?" Additional information documented during the assessment included: location within facility where the participant was working, participant job title, sex, activity performed before the assessment and activity planned to do after the assessment (self-reported). The hand dirtiness participants from baseline and follow-up were not paired. Hand dirtiness assessment results were documented on paper and then transferred to REDCap [6,7].

## In-depth interviews

A convenience sample of one HCW was selected from each HCF to participate in the interview at baseline and follow-up. Questions in baseline interview guides included when and how HCWs practice HH during the workday, HH infrastructure

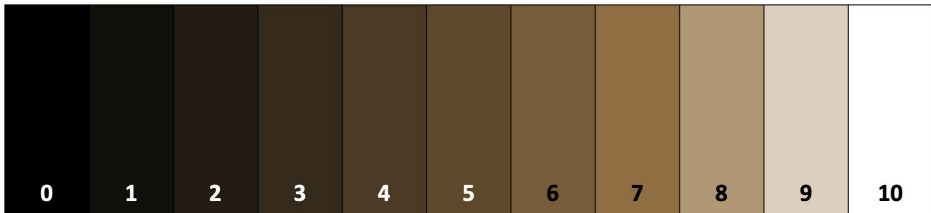

**Fig 2. Hand dirtiness color scale [9].**

and management at the HCF, their perceptions of HH practices, and facilitators and barriers to HH practices at their workplace. Interviews also included facility walk-throughs to visualize the workspace and two card sort activities (Fig 3). In the first card sort activity, participants were asked to rank HH concepts and items that mattered most to them. In the second activity, they were asked to rank how important it is to wash hands in several situations that commonly occur during the workday. Participant rankings along with their reasoning were recorded.

Follow-up interviews were conducted with one HCW who was either a champion or aware of the intervention at each of the 12 pilot HCFs. Follow-up interview participants were asked about their experience with the intervention as a hygiene champion or as a receiver of information from their facility's champion. Questions focused on messages shared, challenges implementing the intervention, feedback from HCWs at the respective HCFs, perceived changes in HH practices of HCWs, and suggestions for improvement of the hand hygiene champion program. Participants were also asked about their current HH practices, any barriers keeping them from practicing HH properly, and recommendations for future HH programs.

Baseline and follow-up interviews were conducted in-person by two Belizean BCM nurses (VR and RB), except for two follow-up interviews that were conducted via audio call by the lead author (ANL), since these interviews were unable to be conducted on the day of the in-person visit. VR and RB had previous experience recruiting and interviewing study participants; ANL had formal training in qualitative research. All interviewers were trained in human subject research and interview techniques. The in-person interview participants were approached and invited to participate in-person. At the

**When you think of hand hygiene at work, what matters most?**

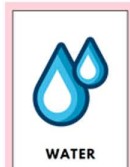 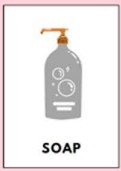 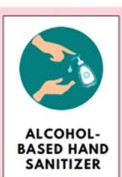 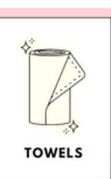 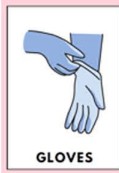 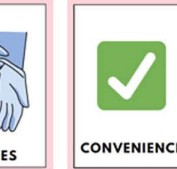 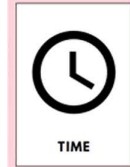 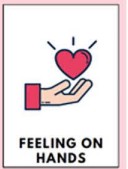 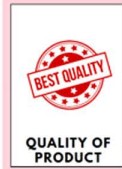

**How important is it to wash hands at work in the following situations?**

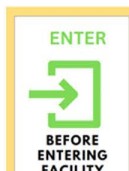 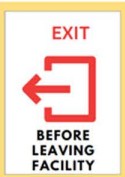 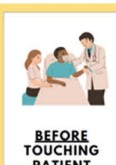 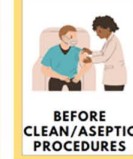 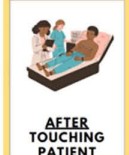 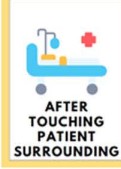 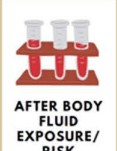 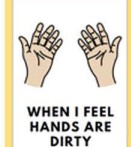 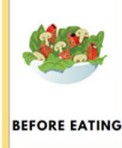

**Fig 3. Cards provided to HCWs for card sort activity.**

facilities where the interviews were not able to be conducted in-person during the follow-up visits, the study team reached out to HCWs they knew were involved with the intervention and invited them to participate in an online interview. The interview guides were reviewed during the training sessions at both study time points, and revisions were made based on feedback from the Belizean enumerators. Participants were informed of the purpose of the interview and verbal consent was obtained from participants prior to the interview and recording. No additional notes were taken during the interview. Interviews were conducted in English; baseline interviews took about 30 minutes and follow-up interviews took about 15 minutes to complete. Interview recordings were transcribed verbatim by a professional transcription service. All transcripts were deidentified prior to analysis and were only accessible to members of the study team. Transcripts were not returned to study participants or any Belizean government official.

## Analysis

Descriptive statistics were calculated to compare HH resources, HHA, and hand dirtiness between baseline and follow-up. Chi-square test was used to assess the significance of the differences between baseline and follow-up. Multilevel logistic regressions were used to assess the association between various exposure variables and two outcome variables: 1) HHA and 2) scoring ≥9 on the hand dirtiness assessment. The cutoff for the hand dirtiness score was selected since approximately half of the scores at baseline and follow-up were 9 and 10. The regressions accounted for potential clustering at the HCF level. The multivariable models included all variables that may be important predictors of HHA and hand dirtiness. The HHA predictors included data collection timepoint, health facility type, healthcare worker role, medical procedures type (healthcare worker was performing during HH observation), moment of contact (before or after patient contact), whether the contact was during a new patient encounter, and HH resources available during the observation. Hand dirtiness predictors included data collection timepoint, health facility type, staff role, and activity HCW was performing before the hand swab. All statistical analyses were performed using STATA version 16 [10]. Statistical tests were considered significant at the 0.05 level.

The interview transcripts were coded using a codebook that included both deductive codes developed using the interview guides and inductive codes from concepts that emerged during thematic analysis. Coding was conducted in MAX-QDA 2020 and 2022 (VERBI Software, 2020/2022) [11]. All transcripts were coded by a primary coder and reviewed by a secondary analyst to ensure codes were applied consistently. After applying the generated codes to a few transcripts, the coding team regrouped to discuss definitions and additional codes needed as common themes emerged. Any changes to code definitions were applied by the secondary analysts. A thematic analysis approach was then used to analyze the baseline and follow-up interviews separately to derive common experiences from the participants.

## Intervention

One of the major findings from the baseline assessments at the 12 pilot HCFs was the gap in HHA during patient care. Due to the limited timeframe of the study, the intervention was designed with the main goal of increasing HH knowledge and practices among HCWs. The HH champions model was selected to complement existing IPC training in Belize and to promote a stronger culture of HH in these HCFs where there is often no designated infection control staff on-site. In August 2023, a one-day in-person workshop was held in Belize City, Belize, to provide an overview of the hand hygiene champion program. The Ministry of Health and Wellness appointed one HCW from each of the selected 12 pilot facilities to participate in the workshop and serve as the champion for their HCF. During the workshop, these selected champions were encouraged to recruit additional colleagues at their HCF to serve as champions.

The champions were asked to promote HH-related themes at their respective facilities, one per week for six weeks (Fig 4). The themes were developed by the research team based on HH motivators shared by HCWs and observed gaps in HH practices from the baseline assessments. The champions were provided with informational materials (flyers, informational sheets, videos, etc.) from the World Health Organization and the U.S. CDC to support their activity development

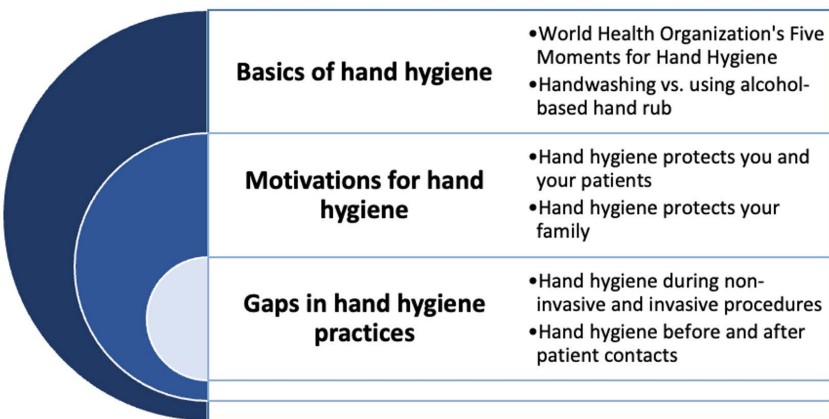

| | |
|---|---|
| **Basics of hand hygiene** | • World Health Organization's Five Moments for Hand Hygiene<br>• Handwashing vs. using alcohol-based hand rub |
| **Motivations for hand hygiene** | • Hand hygiene protects you and your patients<br>• Hand hygiene protects your family |
| **Gaps in hand hygiene practices** | • Hand hygiene during non-invasive and invasive procedures<br>• Hand hygiene before and after patient contacts |

**Fig 4. Hand hygiene champions themes emphasized during the intervention.**

for each week [12,13]. The content of the workshop and the materials provided were reviewed by a group of Belizean nurses before implementation to ensure the context was appropriate for HCFs in Belize. During the workshop, examples were provided to generate ideas for disseminating the weekly themes, but there were no specific criteria for implementation. Champions were encouraged to engage creatively with their staff about the HH topics, and they were asked to share photos and videos of their weekly theme activity implementation with the study team at BCM. The study team was not involved in the implementation of the activities. A suggested schedule was created so that all HCFs would focus on the same themes each week. Fig 5 displays examples of how the program was implemented at two pilot HCFs. During the intervention, champions shared the outlined themes with their colleagues using various methods, including presentations, demonstrations, verbal reminders, and printed flyers. Messages were shared in-person and via group text messages. A few champions extended the intervention to educate patients and other HCWs in nearby facilities.

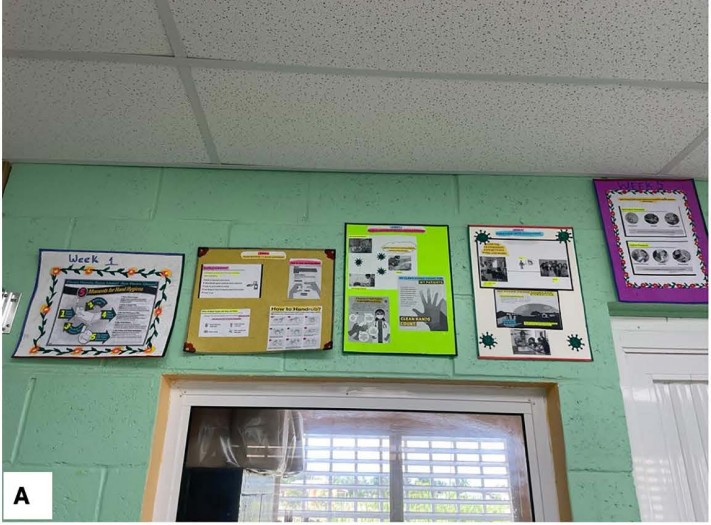
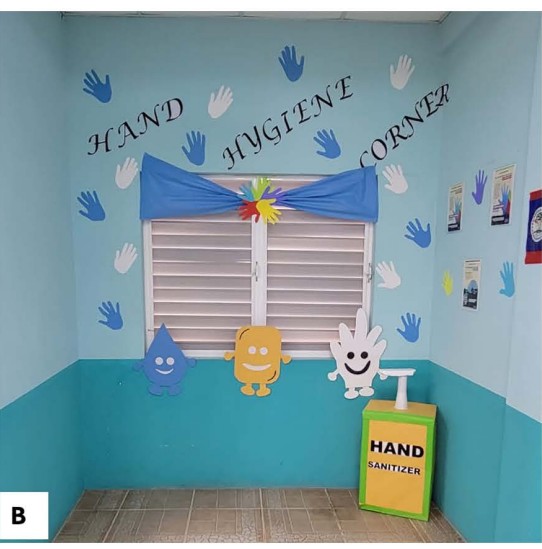

**Fig 5. Examples of program execution of participating facilities – posters for each of the assigned themes (A) and hand hygiene corner to promote hand hygiene awareness among staff and patients (B).**

## Results

Nearly half (42%) of the health centers and polyclinics included in the facility assessment were in the Southern districts of Stann Creek and Toledo (Table 1). Most facilities were classified as health centers (81%). Of the 12 facilities selected for the pilot intervention, 75% were health centers. The average monthly patient volume at the health centers and polyclinics varied widely, ranging from less than 100 outpatient consultations to more than 500.

### Hand hygiene infrastructure and resources

A total of 140 patient care rooms from all 26 HCFs were included in the facility assessment at baseline. Of these rooms, 87 (62%) and 104 (74%) had a handwashing station with soap and water and a functional ABHR dispenser, respectively (Table 2). Most patient care rooms (n = 122, 87%) had either a functional handwashing station with soap or an ABHR dispenser. Among the 12 pilot HCFs, 76 and 69 patient care rooms were assessed at baseline and follow-up, respectively. At baseline at the 12 pilot HCFs, 65 of 76 patient care rooms (86%) had a handwashing station with soap and water or a functional ABHR dispenser compared to 62 of 69 rooms (90%) at follow-up.

All 12 of the baseline interview participants were women, two of whom were physicians, and ten nurses. The 12 follow-up interviews were conducted among 2 male nurses, 7 female nurses, and 3 other HCWs (caregiver, nursing assistant, patient care assistant). Some participants may have been interviewed at baseline and follow-up, but the data were not paired. In baseline interviews, HCWs shared that they had consistent access to handwashing stations with soap or ABHR during the workday. A few HCWs noted that HH can be difficult in some situations due to the placement of handwashing stations or ABHR dispensers (e.g., multiple rooms sharing a handwashing station). Two HCWs reported challenges with water shortages and one HCW mentioned low water pressure. A few HCWs reported having non-functional

**Table 1. Characteristics of polyclinics and health centers included in the study.**

| Characteristics of polyclinics and health centers | All polyclinics and health centers (N = 26) n (%) | Pilot polyclinics and health centers (N = 12) n (%) |
|---|---|---|
| **Region (district)** | | |
| Northern (Corozal and Orange Walk) | 7 (27) | 4 (33) |
| Central (Belize) | 6 (23) | 2 (17) |
| Western (Cayo) | 2 (8) | 2 (17) |
| Southern (Stann Creek and Toledo) | 11 (42) | 4 (33) |
| **Facility type** | | |
| Health center | 21 (81) | 9 (75) |
| Polyclinic | 5 (19) | 3 (25) |
| **Average monthly outpatient consultations** | | |
| < 100 | 6 (23) | 2 (17) |
| 100–249 | 10 (38) | 4 (33) |
| 250–499 | 6 (23) | 3 (25) |
| ≥ 500 | 3 (12) | 3 (25) |
| **Number of clinic staff** | | |
| 1–2 | 8 (31) | 4 (33) |
| 3–5 | 12 (46) | 4 (33) |
| > 5 | 6 (23) | 4 (33) |

Some categories do not add to 100% due to rounding or missing data

**Table 2. Hand hygiene resources in patient care rooms of polyclinics and health centers.**

| Hand hygiene resources | All polyclinics and health centers at baseline n (%) | Pilot polyclinics and health centers n (%) | | | |
| --- | --- | --- | --- | --- | --- |
| | N = 140 | Baseline | Follow-up | Change | |
| | | n (%) | n (%) | % | P-value |
| | | N = 76 | N = 69 | | |
| Rooms with handwashing stations with water | 92 (66) | 51 (67) | 43 (62) | −5 | 0.547 |
| Rooms with handwashing stations with soap and water | 87 (62) | 49 (64) | 41 (59) | −5 | 0.531 |
| Rooms with functional ABHR dispenser* | 104 (74) | 53 (70) | 53 (77) | +7 | 0.403 |
| Rooms with handwashing station with soap and water or functional ABHR dispenser* | 122 (87) | 65 (86) | 62 (90) | +4 | 0.554 |

ABHR = alcohol-based hand rub.

*Functional ABHR dispenser: there is an alcohol-based handrub inside the dispenser and the dispenser is mechanically functional.

handwashing stations or ABHR dispensers. At follow-up, many HCFs continued to have access to HH resources, but inconvenient placement of HH resources remained a challenge for a few HCFs.

"…we always have our dispensers full. We have access to the running water, and soaps, and everything." (Baseline, HCF 4)

"… here in this village, I see a problem with the water system…every day in the morning, we have no water." (Baseline, HCF 8)

"…[the] triage area [doesn't] have water. So, after treating a patient we'll probably have to go to [the] observation room to wash our hands or to the bathroom." (Follow-up, HCF 2)

**Hand hygiene practices**

Hand hygiene observations were conducted at all 12 pilot HCFs at baseline and at 11 HCFs at follow-up since one HCF only had one staff member who was already aware of the purpose of the study. A total of 256 and 258 HH opportunities were observed at baseline and follow-up, respectively. HHA was significantly higher at follow-up (50%) compared to baseline (22%) (aOR = 4.21; 95% CI = 2.70, 6.56) (Table 3). Other factors that were significant predictors of HHA included being an "other" healthcare provider, which included community workers and pharmacists (aOR = 7.64; 95% CI = 1.38, 42.15), invasive procedures (aOR = 1.82; 95% CI = 1.07, 3.09), and performing HH after patient contact (aOR = 1.68; 95% CI = 1.12, 2.52).

At baseline, HCWs washed their hands with soap or used ABHR during 18% of HH opportunities before patient contact and 26% of HH opportunities after patient contact (Fig 6). In comparison, at follow-up, handwashing with soap or ABHR was used during 44% of HH opportunities before patient contact and 55% of HH opportunities after patient contact.

During the card sort activity at baseline, HCWs ranked access to supplies as the factor that "matters most" to them in practicing HH at work; access to water and soap was considered most important, followed by access to ABHR and gloves. Some participants also considered the quality of the HH supplies to be critical. 'Convenience' and 'feelings on hands' were perceived as less important factors for HH practices. Healthcare workers ranked 'before entering the facility' and 'before touching patients' as the most critical moments for HH. Hand hygiene 'before leaving the facility' was consistently ranked as the least important moment. Healthcare workers shared that preventing disease transmission and helping protect those around them as motivators for HH.

**Table 3. Characteristics associated with hand hygiene adherence during patient contact using aggregate data from baseline and follow-up.**

| Variable | Total | HHA (%) | Bivariate | | Multivariable | |
|---|---|---|---|---|---|---|
| | | | OR (95% CI) | P-value | aOR (95% CI) | P-value |
| **Overall** | 514 | 186 (36) | – | – | – | – |
| **Timepoint** | | | | | | |
| Baseline | 256 | 57 (22) | Ref | Ref | Ref | Ref |
| Follow-up | 258 | 129 (50) | 4.08 (2.68, 6.21) | **<0.001** | 4.21 (2.70, 6.56) | **<0.001** |
| **Facility type** | | | | | | |
| Health center | 358 | 132 (37) | Ref | Ref | Ref | Ref |
| Polyclinic | 156 | 54 (35) | 0.87 (0.38, 1.99) | 0.742 | 0.73 (0.29, 1.84) | 0.509 |
| **Healthcare worker role** | | | | | | |
| Physician | 128 | 47 (37) | Ref | Ref | Ref | Ref |
| Nurse | 284 | 92 (32) | 0.74 (0.46, 1.18) | 0.207 | 0.77 (0.45, 1.31) | 0.335 |
| Caretaker/patient care assistant | 88 | 35 (40) | 0.98 (0.51, 1.86) | 0.948 | 1.57 (0.73, 3.39) | 0.252 |
| Other** | 14 | 12 (86) | 7.03 (1.40, 35.24) | **0.018** | 7.64 (1.38, 42.15) | **0.020** |
| **Procedure type** | | | | | | |
| Non-invasive | 402 | 140 (35) | Ref | Ref | Ref | Ref |
| Invasive | 110 | 45 (41) | 1.41 (0.90, 2.22) | 0.13 | 1.82 (1.07, 3.09) | **0.028** |
| **Moment of contact** | | | | | | |
| Before patient contact | 257 | 81 (32) | Ref | Ref | Ref | Ref |
| After patient contact | 257 | 105 (41) | 1.54 (1.06, 2.24) | **0.023** | 1.68 (1.12, 2.52) | **0.011** |
| **New patient encounter** | | | | | | |
| No | 226 | 74 (33) | Ref | Ref | Ref | Ref |
| Yes | 288 | 112 (39) | 1.14 (0.78, 1.68) | 0.499 | 1.06 (0.69, 1.62) | 0.804 |
| **Hand hygiene resources available** | | | | | | |
| ABHR only | 202 | 73 (36) | 1.21 (0.36, 4.03) | 0.755 | 0.50 (0.15, 1.72) | 0.273 |
| Soap and water only | 20 | 5 (25) | Ref | Ref | Ref | Ref |
| Soap and water and ABHR | 292 | 108 (37) | 1.48 (0.46, 4.77) | 0.512 | 1.08 (0.59, 1.97) | 0.808 |

ABHR = alcohol-based hand rub, HHA = hand hygiene adherence, OR = odds ratio, aOR = adjusted odds ratio, CI = confidence interval, Ref = referent group *HHA: handwashing with soap and water or using ABHR; **Other healthcare worker roles include community health worker and pharmacist.

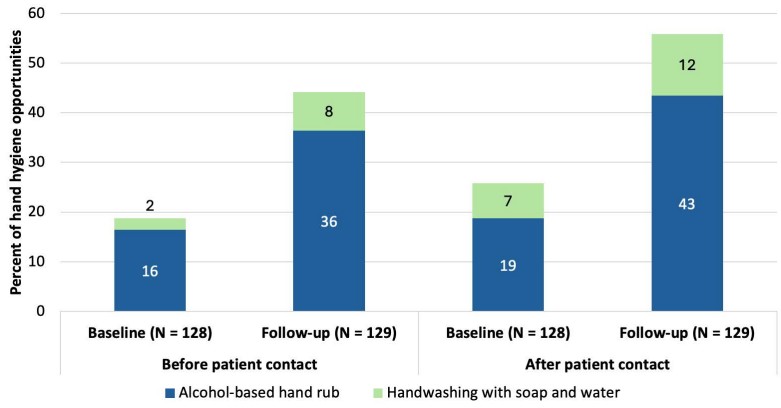

**Fig 6. Hand hygiene methods HCWs practiced before and after patient contact.**

During the baseline and follow-up interviews, HCWs shared that they wash their hands with soap and water and use ABHR at work, but the method used depends on the medical procedures involved and the accessibility of HH resources. Handwashing is performed when invasive procedures are performed or when working with patients with certain signs, such as skin rash or cough. Healthcare workers use ABHR when they need to quickly clean their hands or when there is no access to a handwashing station or water. A few staff mentioned that when they are responding to emergency cases, HH may not be a priority. Some HCWs indicated a preference for hand washing with soap instead of using ABHR. Participants at baseline shared that when using ABHR, HCWs prefer to use liquid ABHR instead of gel-based ABHR to avoid the sticky residue. Some participants at follow-up also mentioned experiencing the sticky feeling from the ABHR. A few staff at follow-up mentioned that they wash their hands after multiple uses of ABHR. A few staff members shared that the COVID-19 pandemic made them more conscious of HH; however, they felt that HH practices were already a habit for HCWs. Some HCWs perceived that HH has improved at their HCF since the intervention.

*"… the best is the handwashing with soap and water, and the hand sanitizer is just optional whenever there is no other way of having the soap and water available..." (Baseline, HCF 7)*

*"…I wash my hands when I do like a really invasive procedure like today, I did a Pap smear. I wash my hands. Or when you feel your hand is dirty after using frequent hand sanitizer then I wash my hands." (Follow-up, HCF 5)*

*"… since COVID, we learned that every time you do something, you sanitize. You wash your hands. I think it's something that you already have. It just clicks, you touch something you wash your hand. (Follow-up, HCF 8)*

**Hand dirtiness**

The median hand dirtiness score at baseline was 8 (range: 3–10) compared to 9 (range: 2–10) at follow-up (Fig 7). Note that a higher score corresponds to less visible debris at the time of the assessment. Furthermore, the visibility of debris is not a measure of the presence of disease-causing pathogens on hands. At baseline, approximately half of HCWs had a hand dirtiness score below 9 compared with 44% of HCWs at follow-up (Table 4). In the bivariate analysis, nurses had significantly higher odds of scoring ≥9 compared to physicians (OR = 3.26; 95% = 1.03, 10.32). When asked about the appropriate HH method for when hands are visibly dirty, (66/91) 73% responded with the correct answer of soap and

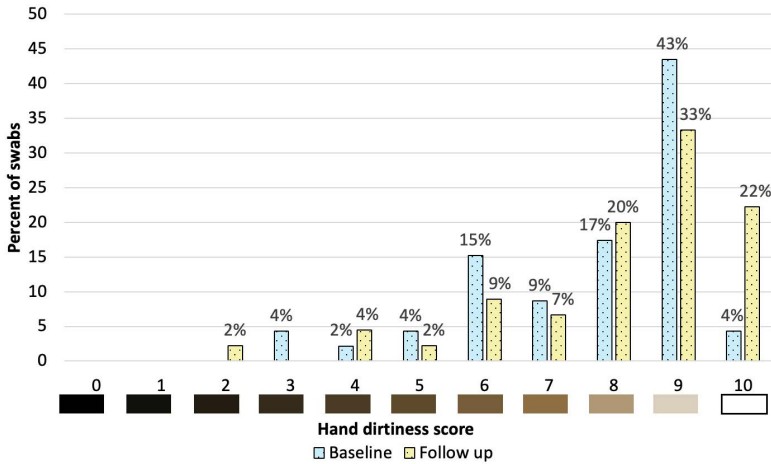

**Fig 7. Distribution of hand dirtiness scores at baseline and follow-up.**

**Table 4. Characteristics associated with having a hand swab score of 9 or 10.**

| Variable | Total | Proportion with hand dirtiness score ≥ 9 n (%) | Bivariate | | Multivariable | |
|---|---|---|---|---|---|---|
| | | | OR (95% CI) | P-value | aOR (95% CI) | P-value |
| **Overall** | 91 | 47 (52) | | | | |
| **Timepoint** | | | | | | |
| Baseline | 46 | 22 (48) | Ref | Ref | Ref | Ref |
| Follow-up | 45 | 25 (56) | 1.41 (0.60, 3.31) | 0.436 | 1.37 (0.56, 3.35) | 0.479 |
| **Facility type** | | | | | | |
| Polyclinic | 29 | 15 (52) | Ref | Ref | Ref | Ref |
| Health center | 62 | 32 (52) | 0.99 (0.32, 3.10) | 0.986 | 0.97 (0.26, 3.63) | 0.969 |
| **Staff role** | | | | | | |
| Physician | 21 | 8 (38) | Ref | Ref | Ref | Ref |
| Nurse | 47 | 30 (64) | 3.26 (1.03, 10.32) | **0.045** | 3.21 (1.00, 10.29) | 0.050 |
| Other | 23 | 9 (39) | 1.17 (0.31, 4.38) | 0.814 | 1.11 (0.29, 4.33) | 0.878 |
| **Sex** | | | | | | |
| Female | 80 | 42 (53) | Ref | Ref | | |
| Male | 11 | 5 (45) | 0.76 (0.21, 2.83) | 0.687 | | |
| **Activity before swab** | | | | | | |
| With patient | 45 | 23 (51) | Ref | Ref | Ref | Ref |
| Not with patient | 46 | 24 (52) | 1.03 (0.44, 2.43) | 0.951 | 1.15 (0.45, 2.90) | 0.773 |
| **Activity after swab** | | | | | | |
| With patient | 36 | 16 (44) | Ref | Ref | | |
| Not with patient | 55 | 31 (56) | 1.65 (0.67, 4.05) | 0.274 | | |
| **When your hands are visibly dirty, should you clean them with soap and water, alcohol-based hand sanitizer, or are both equally effective?** | | | | | | |
| Soap and water | 66 | 34 (52) | Ref | Ref | | |
| ABHR | – | – | – | – | | |
| Both | 25 | 13 (52) | 1.07 (0.40, 2.85) | 0.898 | | |

ABHR = alcohol-based hand rub, OR = odds ratio, aOR = adjusted odds ratio, CI = confidence interval, Ref = referent group, Other staff roles include caretakers, community health workers, patient care assistants, and pharmacists. *A score of 9 was selected as the cutoff since it was the overall median of the baseline and follow-up scores combined. This score does not necessarily indicate the appropriate level of hand cleanliness.

water while the remaining (25/91) 27% said that both soap and water and ABHR are equally effective. No one responded that ABHR alone is effective when hands are visibly dirty.

## Perceptions of intervention

Healthcare workers viewed the program, especially the workshop, as a helpful refresher and expressed interest in implementing future hand hygiene champion programs. Healthcare workers commonly reported that lack of time during the workday and small staff size as barriers to implementing the program activities. Staff from two HCFs expressed that it was difficult to coordinate the intervention since the intervention period coincided with their vacation period. Another champion shared challenges implementing the program since she does not work at the designated HCF daily. A few facilities also had problems accessing supplies to implement the program activities.

  *"Every week, we would do like a …get together...[in] the evenings or at the start of the mornings. And I would share with them different… information about hand hygiene." (Follow-up, HCF 3)*

*"…we don't have a full staff, and it's not something that I can do all the activities the way I would want to do something because I have limited time. And the time that I have, I have to more focus on my patients." (Follow-up, HCF 7)*

### Recommendations from healthcare workers

To improve HH in public polyclinics and health centers, HCWs recommended additional training and collaboration with the Ministry of Health and Wellness. Other suggestions included incentives for staff, funding for program implementation, and a national handwashing day celebration. For the hand hygiene champion program specifically, HCWs recommended direct training with multiple staff at each facility, more frequent follow-up, and provision of additional resources for program implementation. A few facilities suggested including patients in the intervention.

*"…hopefully, future projects...do not take too long in between. Like, after this one ends, hopefully the next project isn't years after. I know there's a lot of projects on the table and so forth. But I feel like hand hygiene is very important in the health system." (Follow-up, HCF 12)*

*"Maybe more frequent visits. You know just to do, like, a follow up. A check-up just to help with more reassurance and probably build that confidence not only in the [Hand Hygiene] Champion representative, but in the entire staff. So that they can feel more…involved and… probably take the program more seriously." (Follow-up, HCF 9)*

## Discussion

This study aimed to provide a cross-sectional evaluation of access to HH infrastructure and resources at 26 public polyclinics and health centers across all six districts of Belize. We also aimed to document changes in HH practices and hand dirtiness among HCWs before and after a behavior change intervention at a subset of 12 pilot facilities. Our assessments showed that HH resources were available in most patient care areas and the availability of HH resources did not vary significantly between baseline and follow-up at the pilot HCFs. Healthcare workers shared positive attitudes towards the hand hygiene champion intervention but expressed challenges in the implementation. There was a significant increase in HHA among HCWs following the intervention; however, there were remaining gaps in HHA and hand dirtiness.

A high proportion of patient care rooms were in proximity to a handwashing station with soap or a functional ABHR dispenser, with most of the rooms having a functional ABHR dispenser. Although many of these health centers and polyclinics are in rural areas, there is a formal system for requesting ABHR from nearby HCFs or central pharmacies where ABHR is locally produced, which may have contributed to the constant availability of supplies. Despite the presence of HH supplies, there were still gaps in HH practices after the intervention. As noted during the interviews, the placement of supplies may be a barrier to HH practices. Healthcare workers shared during the interviews that hygiene may be overlooked in instances of high patient flow or emergency situations, especially if the HH station is in another room within the facility. These experiences are consistent with other studies where lack of time, high workload, and inconvenient placements of HH stations were documented as challenges to HHA [14,15]. A previous qualitative study in HCFs in Belize revealed similar challenges [16]. Many Belizean HCWs perceived handwashing with soap and water as the preferred HH method. However, ABHR is the preferred HH method in most healthcare settings and could be a quicker and more convenient solution for when there is a high patient volume [17]. Future HH promotion programs could consider additional sensitization on the benefits of ABHR in clinical settings and exploring placements of ABHR stations at more convenient point of care locations.

Although HCWs shared positive views on the hand hygiene champion intervention, the main challenges with program implementation were small staff and lack of time during the workday. An ethnographic study in Veteran Affairs hospitals in the U.S. also found that competing priorities during the workday and the turnover or shortages of staff can be barriers

to a hand hygiene champion program [18]. This suggests that the hand hygiene champion model may need to be tailored to accommodate smaller clinical sites. For example, the intervention could facilitate the collaboration of clusters of HCFs instead of individual facilities, with the champions circulating several HCFs. Furthermore, future programs could explore how existing designated IPC staff assigned throughout each district could assist with the program implementation. Nevertheless, there was a significant improvement in HHA from baseline to follow-up. Similarly, a Dutch study using team and leader-directed strategies to promote HH observed improvement in adherence (20% to 53%) among HCWs [19]. A role model intervention in a hospital pediatrics department in Indonesia documented an increase in HHA from 24.1% pre-intervention to 43.7% post-intervention [20]. Another HH promotion intervention in a hospital setting in Malaysia observed improvement in HHA when using peer-identified change agents and management-selected change agents from 48% to 58% and 50% to 64%, respectively [21]. Although we observed an increase in HHA in our study, HCWs still only practiced proper HH during half of the HH opportunities at follow-up. This suggests that continuous intervention and monitoring are necessary to identify and reduce the gaps in HHA in health centers and polyclinics in Belize. Future programs could consider multiple rounds of evaluation to measure long-term and sustained impact of the intervention. Additionally, there is a need to consider how the champion program could be integrated into existing IPC strategies in healthcare facilities in Belize.

The hand dirtiness assessment showed that there was a high proportion of HCWs at baseline and follow-up with visible debris on their hands. We found nurses to have higher odds of scoring ≥9 on the hand dirtiness assessment (having less visible debris) compared to physicians. However, this does not necessarily correlate with HHA. During our direct observations, we did not detect a significant difference in HHA between physicians and nurses.

During the in-depth interviews, staff perceived that HH has become a norm since the start of the COVID-19 pandemic and they self-reported practicing HH before and after patient contacts and after performing certain invasive procedures. However, our direct observation of HH practices showed that HHA was higher after patient contact compared with before patient contact. This is consistent with the literature on HHA in HCFs, including a previous study in Belizean hospitals and polyclinics [22]. A meta-analysis on HH among HCWs during the COVID-19 pandemic showed that adherence was highest after contact with patient's bodily fluid followed by after direct contact with the patient and was lowest before patient contact [3]. It is interesting to note that during our card sort activity, HCWs ranked before patient contact as one of the most important situations for HH, but this was not observed during actual practices. This suggests that HH knowledge and perceptions may not result in behavior adoption. We also observed that HHA was higher during invasive procedures, which may be because these procedures act as a visual cue for HH. Additionally, HCWs may have higher perceived risk of disease transmission during these instances compared to when performing non-invasive procedures. A study in a tertiary hospital in Spain found that HHA was higher during situations where HCWs perceived a high risk for infection [23]. The WHO has outlined a multimodal hand hygiene improvement strategy including five essential elements: system change, training/education, evaluation and feedback, reminders, and institutional safety climate [24]. To complement existing HH and IPC educational programs in Belize, a multimodal intervention with these other elements could help reduce the gaps in HHA during critical moments [2]. In addition to trainings, direct observations and constructive feedback to HCWs may be helpful to close the gaps during patient care.

This study is subject to several limitations. Due to the short timeframe of the study, the follow-up assessments were conducted immediately after the intervention, which did not allow the study to evaluate the long-term impact and the sustainability of the intervention. The HHA observed at follow-up may be inflated due to the short gap between the intervention and the evaluation. Additionally, HHA may have been positively skewed at follow-up since HCWs were then aware of the objective of the project and may have modified their behavior during HH observations. Additionally, since the observations were only conducted before and after patient contacts, some of the WHO's five moments may have been missed. The interpretation of HHA in this study are limited to the observations before and after patient contacts. Future studies could consider expanding observations to include all five moments of HH. The impact of the intervention may have been

limited by the low fidelity of the intervention implementation. Although the hand hygiene champion program was designed to be a six-week intervention, some facilities had a late start due to logistical challenges; therefore, these facilities were instructed to cover multiple themes in one week. Another facility could not complete the program due to construction at the health center. Furthermore, one facility was staffed by one person, therefore, the intervention was not able to be executed (no HH observations were conducted at this HCF at follow-up). Although weekly reminders were sent each week, themes may have been skipped at some facilities. Since not all HCFs reported their progress weekly, there was no comprehensive documentation of if and how the themes were implemented across the pilot HCFs. Between baseline and follow-up, there were changes in staff at several facilities. As a result, there may be changes to HH practices from baseline to follow-up that may have occurred regardless of the intervention. Lastly, since the data at baseline and follow-up were not paired, the same HCWs may have participated both at baseline and follow-up; thus, the independent observation assumption may be violated. Although qPHAT is a validated instrument to measure hand dirtiness, there remain challenges to the interpretation of the data since no specific score indicates appropriate cleanliness nor the presence of disease-causing pathogens. Additionally, qPHAT was not designed specifically to measure hand dirtiness in healthcare facilities and to our knowledge, no validation tests had been conducted in healthcare facilities in Central America. Lastly, the selected HCFs may not be representative of polyclinics and health centers in the country.

Despite these limitations, this study was the first to systematically assess the HH infrastructure and resources nationally at polyclinics and health centers in Belize and it showed that the hand hygiene champion model may have been effective at improving HH as demonstrated through the increase in HHA and positive HH perceptions. Additionally, the study utilized a mixed-methods approach to evaluating the impact of a behavior change intervention on HH practices among HCWs. The use of multiple assessment tools allowed us to capture robust information to compare HCWs' perceptions and documented HH resources and practices. The feedback from the intervention is beneficial for future implementation of HH programs in HCFs in Belize and similar settings.

## Conclusion

A peer-led program like the piloted HH champions intervention may be an effective approach to fostering a culture of HH among HCWs. Future HH programs may consider tailoring the intervention based on HCWs' feedback to meet the needs of these smaller HCFs to increase intervention fidelity and impact. Despite the improvement in HHA, there were remaining gaps, suggesting a need for a comprehensive intervention including multiple components of the WHO multimodal hand hygiene strategy as part of infection prevention and control programs.

## Acknowledgments

We would like to thank the HCWs at the HCFs for their support and participation. We would also like to acknowledge the contribution of Mr. Nicholas Bivens and Dr. Alexandra Medley for their contribution to the study protocol and data collection instruments from a previous project, which were adapted and used for this study.

The findings and conclusions of this report are those of the authors and do not necessarily represent the official position of the U.S. Centers for Disease Control (CDC).

## Author contributions

**Conceptualization:** Anh N. Ly, Kelsey McDavid, Christina Craig, Melissa Diaz-Musa, Francis Morey, Russell Manzanero, Gerhaldine Morazan, Matthew Lozier, Kristy O. Murray.

**Data curation:** Anh N. Ly.

**Formal analysis:** Anh N. Ly, Makenzie Towery, Anna Impellitteri.

**Funding acquisition:** Kristy O. Murray.

**Investigation:** Anh N. Ly, Rosalva Blanco, Vickie Romero, Gerhaldine Morazan.

**Methodology:** Anh N. Ly, Kelsey McDavid, Christina Craig, Melissa Diaz-Musa, Francis Morey, Russell Manzanero, Gerhaldine Morazan, Matthew Lozier, Kristy O. Murray.

**Project administration:** Anh N. Ly.

**Software:** Anh N. Ly.

**Supervision:** Anh N. Ly, Matthew Lozier, Kristy O. Murray.

**Writing – original draft:** Anh N. Ly.

**Writing – review & editing:** Anh N. Ly, Kelsey McDavid, Christina Craig, Matthew Lozier, Kristy O. Murray.

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
