## [Decision Letter · Decision Letter 0]

12 Aug 2025

PONE-D-25-35439
Mixed-methods evaluation and behavior change interventions to improve hand hygiene resources and practices among healthcare workers in rural polyclinics and health centers in Belize, 2023
PLOS ONE

Dear Dr. Murray,

Thank you for submitting your manuscript to PLOS ONE. After careful consideration, we feel that it has merit but does not fully meet PLOS ONE’s publication criteria as it currently stands. Therefore, we invite you to submit a revised version of the manuscript that addresses the points raised during the review process.
 
Both reviewers have some comments that need addressing, particularly around making the limitations clear. 

We look forward to receiving your revised manuscript.

Kind regards,

Alison Parker

Academic Editor

PLOS ONE

2. In the ethics statement in the Methods, you have specified that verbal consent was obtained. Please provide additional details regarding how this consent was documented and witnessed, and state whether this was approved by the IRB.

Additional Editor Comments (if provided):

Reviewers' comments:

Reviewer's Responses to Questions

**Comments to the Author**

1. Is the manuscript technically sound, and do the data support the conclusions?

Reviewer #1: Partly

Reviewer #2: Yes

2. Has the statistical analysis been performed appropriately and rigorously?

Reviewer #1: I Don't Know

Reviewer #2: Yes

3. Have the authors made all data underlying the findings in their manuscript fully available?

Reviewer #1: No

Reviewer #2: Yes

4. Is the manuscript presented in an intelligible fashion and written in standard English?

Reviewer #1: Yes

Reviewer #2: Yes

5. Review Comments to the Author

Reviewer #1: Thank you to the authors for their work. Overall, this study provides reference value for the field of hand hygiene. However, I have several suggestions:

The methods section of the abstract needs to clearly specify the specific intervention measures and statistical analysis methods; the conclusion section of the abstract needs to be more specific, as the specific intervention measures cannot be determined from the entire abstract.

The introduction should include more details about the specific situation of hand hygiene in Belize.

Methods section: The title of the paper clearly limits the region to rural areas, but from the description in the methods section, it cannot be determined that the 26 sites and the subsequent 12 sites belong to rural areas; more information about the sites needs to be added, such as whether the department settings are consistent, the educational level of medical staff, etc.; the variables of logistic regression and their definitions should be listed; it is recommended to provide the source code and source data.

There are deficiencies in representativeness: the inclusion and exclusion criteria for the sites are open to discussion. The selection basis for the 26 sites is not objective enough, and the subsequent selection of 12 sites based on the number of people is not representative.

The description of intervention measures needs to be more specific, as the specific steps of the intervention measures cannot be determined from the methods section.

The methods section should further clarify, after selecting the sites, the basis for selecting the respondents, the number of respondents, and the number of observations (it is currently unclear how many times the same person was observed and whether different people were observed the same number of times).

The discussion section should explore more relevant studies in the country

Check whether the references actually exist.

Reviewer #2: I enjoyed reading this manuscript. It’s a clear and thoughtfully designed mixed-methods study that looks at both the availability of hand hygiene (HH) resources and the impact of a peer-led champion intervention in rural Belizean health facilities. This focus on small outpatient settings in a low-resource context is important and often overlooked in infection prevention and control (IPC) research. The mixed approach of combining quantitative data with in-depth interviews makes the findings richer and more meaningful.

The work is technically sound. The data are collected systematically, the statistical analysis is appropriate for the study design, and the results are interpreted sensibly, with limitations openly acknowledged. The paper is well structured, easy to follow, and written in clear English.

Major Points to Think About

Timing of Follow-Up

The follow-up was done right after the six-week program. This makes it hard to know whether the improvements will stick over time and could mean staff were on their “best behavior” because they knew they were being observed. It would be good to expand on this in the discussion and talk about how future work might check for longer-term impact.

How Sites Were Chosen

Choosing pilot sites based on staff size is practical, but it might mean the results lean toward better-staffed, better-resourced facilities. A short note on how this could affect generalizability would be helpful.

What Was Observed

HH adherence was only tracked before and after patient contact, so some of the WHO “5 Moments” were missed. Explaining why this approach was taken and what that means for interpreting the results would add clarity.

Hand Dirtiness Scores

The qPHAT tool measures visible dirt, not germs. You’ve already noted this, but I’d highlight it a bit more so readers don’t confuse “clean-looking” with “pathogen-free.”

Minor Suggestions

The qualitative results are engaging. Linking the barriers staff mentioned more directly to specific, actionable program improvements would make them even more useful.

It would be great to say a little more about how this champion model could fit into Belize’s national IPC strategies for long-term, sustainable scale-up.

Overall Impression

This is a solid, relevant, and well-written piece of work that has real-world value for IPC programs in similar contexts. The recommendations are practical, and the mix of methods gives the findings both depth and credibility. With a few clarifications and some extra discussion on sustainability and generalizability, it will be even stronger.

6. PLOS authors have the option to publish the peer review history of their article (what does this mean?). If published, this will include your full peer review and any attached files.

Reviewer #1: No

Reviewer #2: No

---

## [Author Response · Author response to Decision Letter 1]

16 Sep 2025

Thank you for the opportunity to resubmit. The comments and suggestions from the reviewers were very helpful. Our specific responses are below:

Reviewer #1: Thank you to the authors for their work. Overall, this study provides reference value for the field of hand hygiene. However, I have several suggestions:

The methods section of the abstract needs to clearly specify the specific intervention measures and statistical analysis methods; the conclusion section of the abstract needs to be more specific, as the specific intervention measures cannot be determined from the entire abstract.

Thank you for the suggestion. We edited the methods section of the abstract for clarity on the intervention and analytical approaches (lines 34-38).

The introduction should include more details about the specific situation of hand hygiene in Belize.

There is currently limited HH data in healthcare settings in Belize, especially in outpatient clinics. We have added a sentence about the gaps in HH practices as observed in our previous study in large hospitals and polyclinics. (line 28).

Methods section: The title of the paper clearly limits the region to rural areas, but from the description in the methods section, it cannot be determined that the 26 sites and the subsequent 12 sites belong to rural areas; more information about the sites needs to be added, such as whether the department settings are consistent, the educational level of medical staff, etc.; the variables of logistic regression and their definitions should be listed; it is recommended to provide the source code and source data.

We have removed “rural” from the title to be more inclusive of geographic locations. We do not have information on educational level of medical staff; therefore, that information was not included. We have added details about the predictors included in the logistic regression models (lines 206-211).

There are deficiencies in representativeness: the inclusion and exclusion criteria for the sites are open to discussion. The selection basis for the 26 sites is not objective enough, and the subsequent selection of 12 sites based on the number of people is not representative.

The 26 health facilities included every possible polyclinic and health centers in the country that were operational and not already included in our previous hand hygiene study. We have added a sentence to the limitation section about the limited representativeness of the 12 pilot HCFs (line 480).

The description of intervention measures needs to be more specific, as the specific steps of the intervention measures cannot be determined from the methods section.

Please refer to the intervention section in the methods section for details (line 222).

The methods section should further clarify, after selecting the sites, the basis for selecting the respondents, the number of respondents, and the number of observations (it is currently unclear how many times the same person was observed and whether different people were observed the same number of times).

The “hand hygiene observations” section under methods listed the selection method (random selection), the number of respondents per HCF (three HCWs), and the number of observations (five patient contacts or up to one hour of observation) (line 119).

The discussion section should explore more relevant studies in the country.

We have added results from our quantitative and qualitative evaluations in Belizean hospitals and polyclinics (lines 414 and 444).

Check whether the references actually exist.

Reviewer #2: I enjoyed reading this manuscript. It’s a clear and thoughtfully designed mixed-methods study that looks at both the availability of hand hygiene (HH) resources and the impact of a peer-led champion intervention in rural Belizean health facilities. This focus on small outpatient settings in a low-resource context is important and often overlooked in infection prevention and control (IPC) research. The mixed approach of combining quantitative data with in-depth interviews makes the findings richer and more meaningful.

The work is technically sound. The data are collected systematically, the statistical analysis is appropriate for the study design, and the results are interpreted sensibly, with limitations openly acknowledged. The paper is well structured, easy to follow, and written in clear English.

Major Points to Think About

Timing of Follow-Up

The follow-up was done right after the six-week program. This makes it hard to know whether the improvements will stick over time and could mean staff were on their “best behavior” because they knew they were being observed. It would be good to expand on this in the discussion and talk about how future work might check for longer-term impact.

We have added a sentence to the discussion suggesting future programs to consider long-term evaluations.

How Sites Were Chosen

Choosing pilot sites based on staff size is practical, but it might mean the results lean toward better-staffed, better-resourced facilities. A short note on how this could affect generalizability would be helpful.

Thank you for the suggestion. We have added a sentence to the limitations section about the limited generalizability of the 12 pilot HCFs (line 480).

What Was Observed

HH adherence was only tracked before and after patient contact, so some of the WHO “5 Moments” were missed. Explaining why this approach was taken and what that means for interpreting the results would add clarity.

We have added a sentence to the limitations paragraph acknowledging the limited interpretation of hand hygiene adherence and a suggestion for future studies to consider all five moments of hand hygiene (line 468).

Hand Dirtiness Scores

The qPHAT tool measures visible dirt, not germs. You’ve already noted this, but I’d highlight it a bit more so readers don’t confuse “clean-looking” with “pathogen-free.”

We have added a sentence to the results reminding the readers that the visibility of debris is not a measure for the presence of disease-causing pathogens.

Minor Suggestions

The qualitative results are engaging. Linking the barriers staff mentioned more directly to specific, actionable program improvements would make them even more useful.

We have added some suggestions throughout the discussion section.

It would be great to say a little more about how this champion model could fit into Belize’s national IPC strategies for long-term, sustainable scale-up.

As the IPC program in Belize continues to grow, we have added a suggestion to assess how this hand hygiene champion program can be integrated into national strategies.

Overall Impression

This is a solid, relevant, and well-written piece of work that has real-world value for IPC programs in similar contexts. The recommendations are practical, and the mix of methods gives the findings both depth and credibility. With a few clarifications and some extra discussion on sustainability and generalizability, it will be even stronger.

---

## [Decision Letter · Decision Letter 1]

6 Oct 2025

Mixed-methods evaluation and behavior change interventions to improve hand hygiene resources and practices among healthcare workers in rural polyclinics and health centers in Belize, 2023

PONE-D-25-35439R1

Dear Dr. Murray,

We’re pleased to inform you that your manuscript has been judged scientifically suitable for publication and will be formally accepted for publication once it meets all outstanding technical requirements.

Kind regards,

Alison Parker

Academic Editor

PLOS ONE

Additional Editor Comments (optional):

Reviewers' comments:

Reviewer's Responses to Questions

**Comments to the Author**

1. If the authors have adequately addressed your comments raised in a previous round of review and you feel that this manuscript is now acceptable for publication, you may indicate that here to bypass the “Comments to the Author” section, enter your conflict of interest statement in the “Confidential to Editor” section, and submit your "Accept" recommendation.

Reviewer #2: All comments have been addressed

2. Is the manuscript technically sound, and do the data support the conclusions?

Reviewer #2: Yes

3. Has the statistical analysis been performed appropriately and rigorously?

Reviewer #2: Yes

4. Have the authors made all data underlying the findings in their manuscript fully available?

Reviewer #2: No

5. Is the manuscript presented in an intelligible fashion and written in standard English?

Reviewer #2: Yes

6. Review Comments to the Author

Reviewer #2: The revised manuscript is methodologically sound, clearly written, and addresses the prior reviewer comments thoroughly. The study fits well within PLOS ONE’s scope (public health, WASH, behavior change, applied implementation science), and it contributes useful evidence from an under-researched setting (Belize).

7. PLOS authors have the option to publish the peer review history of their article (what does this mean?). If published, this will include your full peer review and any attached files.

Reviewer #2: No

---

## [Editor Report · Acceptance letter]

PONE-D-25-35439R1

PLOS ONE

Dear Dr. Murray,

I'm pleased to inform you that your manuscript has been deemed suitable for publication in PLOS ONE. Congratulations! Your manuscript is now being handed over to our production team.

Kind regards,

on behalf of

Dr. Alison Parker

Academic Editor

PLOS ONE